# Potential of Coffee Cherry Pulp Extract against Polycyclic Aromatic Hydrocarbons in Air Pollution Induced Inflammation and Oxidative Stress for Topical Applications

**DOI:** 10.3390/ijms25179416

**Published:** 2024-08-30

**Authors:** Weeraya Preedalikit, Chuda Chittasupho, Pimporn Leelapornpisid, Natthachai Duangnin, Kanokwan Kiattisin

**Affiliations:** 1Department of Pharmaceutical Sciences, Faculty of Pharmacy, Chiang Mai University, Chiang Mai 50200, Thailand; weeraya.pr@up.ac.th (W.P.); chuda.c@cmu.ac.th (C.C.); 2Department of Cosmetic Sciences, School of Pharmaceutical Sciences, University of Phayao, Phayao 56000, Thailand; 3Faculty of Pharmacy, Chiang Mai University, Chiang Mai 50200, Thailand; pimporn.lee@cmu.ac.th; 4Regional Medical Science Center 1 Chiang Mai, Chiang Mai 50180, Thailand; natthachai.d@dmsc.mail.go.th

**Keywords:** coffee cherry pulp, pollution, polycyclic aromatic hydrocarbons, anti-inflammation, tumor necrosis factor-alpha, interleukin-6, inducible nitric oxide synthase, cyclooxygenase-2, chlorogenic acid, caffeine, theophylline

## Abstract

Airborne particulate matter (PM) contains polycyclic aromatic hydrocarbons (PAHs) as primary toxic components, causing oxidative damage and being associated with various inflammatory skin pathologies such as premature aging, atopic dermatitis, and psoriasis. Coffee cherry pulp (CCS) extract, rich in chlorogenic acid, caffeine, and theophylline, has demonstrated strong antioxidant properties. However, its specific anti-inflammatory effects and ability to protect macrophages against PAH-induced inflammation remain unexplored. Thus, this study aimed to evaluate the anti-inflammatory properties of CCS extract on RAW 264.7 macrophage cells exposed to atmospheric PAHs, compared to chlorogenic acid (CGA), caffeine (CAF), and theophylline (THP) standards. The CCS extract was assessed for its impact on the production of nitric oxide (NO) and expression of tumor necrosis factor-alpha (TNF-*α*), interleukin-6 (IL-6), inducible nitric oxide synthase (iNOS), and cyclooxygenase-2 (COX-2). Results showed that CCS extract exhibited significant antioxidant activities and effectively inhibited protease and lipoxygenase (LOX) activities. The PAH induced the increase in intracellular reactive oxygen species, NO, TNF-*α*, IL-6, iNOS, and COX-2, which were markedly suppressed by CCS extract in a dose-dependent manner, comparable to the effects of chlorogenic acid, caffeine, and theophylline. In conclusion, CCS extract inhibits PAH-induced inflammation by reducing pro-inflammatory cytokines and reactive oxygen species (ROS) production in RAW 264.7 cells. This effect is likely due to the synergistic effects of its bioactive compounds. Chlorogenic acid showed strong antioxidant and anti-inflammatory activities, while caffeine and theophylline enhanced anti-inflammatory activity. CCS extract did not irritate the hen’s egg chorioallantoic membrane. Therefore, CCS extract shows its potential as a promising cosmeceutical ingredient for safely alleviating inflammatory skin diseases caused by air pollution.

## 1. Introduction

Particulate matter (PM) is composed of a broad range of harmful substances, such as nitric oxide, volatile organic compounds, and polycyclic aromatic hydrocarbons (PAHs). PAHs are commonly found in polluted environments and exert adverse effects on human health, including respiratory tract diseases, cardiovascular diseases, and systemic inflammation [1,2]. Among the organs affected, the skin serves as the primary natural defense barrier of the immune system against ambient pollutants, making it susceptible to inflammatory diseases. Macrophages, as immune cells, act as the first line of defense when the skin is exposed to pollutants and primarily release cytokines [3,4]. When PAHs penetrate the skin, they can cause oxidative stress and inflammation by activating the nuclear factor kappa B (NF-*κ*B) signaling pathway. This pathway regulates numerous genes associated with the immune system, leading to increased levels of pro-inflammatory cytokines such as tumor necrosis factor-alpha (TNF-*α*) and interleukin-6 (IL-6) in keratinocytes [5,6]. Moreover, PM has a dose-dependent effect on increasing the levels of other pro-inflammatory mediators, such as nitric oxide (NO) and cyclooxygenase-2 (COX-2), which are involved in inflammatory skin diseases such as premature aging, acne, atopic dermatitis, and psoriasis [7,8,9,10]. Therefore, the suppression of these pro-inflammatory mediators could be an effective strategy for treating inflammatory skin diseases. Researchers have attempted to identify novel biological components from different plant sources that can protect the skin from PM damage with lower toxicity and stronger anti-inflammatory properties for therapeutic use [11,12,13].

Compounds found in nature that possess antioxidant and anti-inflammatory characteristics are considered potential therapeutic agents for alleviating damage caused by pollutants. Coffee cherry pulp (CCS), a by-product of coffee bean harvesting, is typically considered waste but holds significant potential for various health-promoting effects. The extract contains three major compounds: CGA, CAF, and THP, which offer antioxidant, anti-inflammatory, anti-aging, antimicrobial, and hepatoprotective properties [14,15]. CGA has a broad range of potential biological abilities that contribute to many health advantages, including antioxidant, anti-inflammatory, antimicrobial, anticarcinogenic, anti-diabetic, and strategies for obesity [16,17]. CAF exhibits antioxidation, neuroprotective properties, anti-cancer activity, anti-inflammation, and protective effects against cardiovascular and neurological disorders [18,19]. THP has also demonstrated antioxidant and anti-inflammatory properties, indicating its promise in reducing oxidative stress caused by free radicals and preventing the oxidation of lipids [20]. Therefore, the anti-inflammatory effect of coffee cherry and its main compounds has been accepted, considering the potential of phytochemicals to regulate the excessive production of pro-inflammatory cytokines and reactive oxygen species (ROS).

Additionally, our previous studies found that ethanolic coffee cherry pulp extracted by Soxhlet extraction contained three major constituents, including CGA (17.3 ± 0.0 mg/g extract), CAF (45.0 ± 0.8 mg/g extract), and THP (45.9 ± 1.0 mg/g extract). These constituents exhibited strong potential to protect keratinocyte cells from the harmful effects of ambient PAHs by enhancing cellular antioxidant capacity [14]. Despite extensive research on the health benefits of coffee cherry pulp, its biological activities related to skin health, specifically the anti-inflammatory effects on PAH-induced inflammatory responses and safety, have not been thoroughly investigated. Therefore, this study aimed to explore the antioxidant and anti-inflammatory activities of CCS extract on oxidative stress and inflammation induced by atmospheric PAHs in macrophage cells. To avoid overlooking synergistic effects, this study emphasized the importance of examining the whole CCS extract rather than focusing solely on individual compounds. The effects of CCS extract were compared to those of its standard compounds, including chlorogenic acid, caffeine, and theophylline, for potential use in the cosmetic/cosmeceutical field. To evaluate the potential of CCS extract, we examined its impact on cell viability, ROS production, and the expression of pro-inflammatory cytokines (TNF-*α*, IL-6) and enzymes (iNOS and COX-2) in RAW 264.7 cells exposed to PAHs. Additionally, the hen’s egg test–chorioallantoic membrane (HET-CAM) assay was used to confirm the safety of CCS extract for topical application.

## 2. Results and Discussion

### 2.1. Antioxidant Activities

The ABTS and FRAP assays, methods developed to simulate the radical scavenging process, were used to evaluate the antioxidant activity of CCS extract compared to CGA, CAF and THP. The ABTS assay measures the ability of antioxidants to scavenge ABTS radicals, while the FRAP assay assesses antioxidants’ ability to reduce Fe^3+^ to Fe^2+^, indicating their reducing power [21,22]. The results are shown in Table 1. The ABTS assay results indicated that CGA (310.3 ± 2.5 µM Trolox/g) exhibited the strongest radical scavenging activity, followed by CAF (213.8 ± 1.0 µM Trolox/g) and THP (204.6 ± 1.4 µM Trolox/g). Similarly, CGA exhibited the highest reducing properties, followed by CAF and THP, with FRAP values of 18.71 ± 1.3 mM FeSO_4_, 10.05 ± 0.9 mM FeSO_4_, and 8.30 ± 1.0 mM FeSO_4_, respectively. The presence and positioning of hydroxyl groups in CGA allow it to donate hydrogen atoms to neutralize free radicals and reduce oxidizing agents, making it a more effective antioxidant compared to CAF and THP [23]. Moreover, the relatively lower activity of CAF and THP in both tests highlights the superior capacity of CGA to scavenge radicals within the extract.

Although the radical scavenging and reducing abilities of the CCS extract were not as potent as those of CGA, the extract still demonstrated strong antioxidant activity. It effectively scavenged ABTS radicals with an activity of 193.6 ± 3.7 µM Trolox/g and exhibited a FRAP value of 6.52 ± 0.7 mM FeSO_4_. These data support the notion that the whole crude extract, with its multiple bioactive compounds, particularly CGA, contributes to its overall antioxidant capacity more effectively than individual compounds [24].

### 2.2. Lipoxygenase and Protease Inhibitory Effects

Lipoxygenases (LOXs) are enzymes that convert arachidonic, linoleic, and other polyunsaturated fatty acids into physiologically active metabolites, which play roles in inflammatory and immune responses and serve as sources of ROS in the body [25,26]. LOXs and their signaling pathways are also involved in the pathogenesis of several diseases, including asthma, ulcerative colitis, and psoriasis [27]. Protease, which catalyzes the hydrolysis of peptide bonds to break down proteins, includes several types, such as metalloproteases (MMPs) [28]. Therefore, inhibiting protease can prevent protein denaturation and potentially reduce the onset of inflammation.

This study investigated the inhibitory effects of CCS extract compared to CGA, CAF, and THP on LOX and protease enzymes, which are crucial in inflammatory processes. Given CCS extract’s known free radical scavenging ability, it may serve as an effective inhibitor of LOX and protease enzymes. The findings revealed that CGA and THP (1000 µg/mL) were highly effective at inhibiting protease activity, with inhibition rates of 86.8% and 84.9%, respectively. These results were comparable to the positive control, diclofenac sodium, which exhibited a 95.6% inhibition rate. CAF also demonstrated substantial inhibition at 75.8% (Figure 1b). Although the protease inhibition of CCS extract was less effective, it still showed promising activity with a 69.5% inhibition rate at a concentration of 1000 µg/mL. Interestingly, CCS extract exhibited comparable LOX inhibition to the individual standard compounds, with an inhibition rate of 39.0% (Figure 1a). These results are consistent with previous studies highlighting natural polyphenols as potent inhibitors of LOX [29,30,31]. Cao et al. demonstrated the strong inhibitory effects of CGA on LOX through molecular docking analysis [32]. Additionally, crude extracts from the fruit and leaves of *Zanthoxylum armatum*, containing various alkaloids, showed significant enzyme inhibition with lower IC_50_ values of 21 and 62 μg/mL against LOX [33]. The multiple hydroxyl groups in CGA enhance its inhibitory effects on protease enzymes, specifically matrix metalloprotease-9 (MMP-9) [34]. These data suggest that the strong enzyme inhibitory activities of CCS extract could be attributed to the synergistic interactions of the phytochemicals identified in the extract.

### 2.3. Anti-Inflammatory Effect of Coffee Cherry Pulp Extract, Chlorogenic Acid, Caffeine, and Theophylline against PAH-Induced Oxidative Stress and Inflammation

#### 2.3.1. Effect on the Viability of RAW 264.7 Cells Exposed to PAHs

Humans are commonly exposed to a mixture of PAHs in the air rather than a single PAH component. Combination toxicity typically occurs, where certain PAHs can have either synergistic or antagonistic effects [35]. The MTT test was conducted to assess the cytotoxic effects of PAHs on RAW 264.7 cells. The cells were exposed to different doses of PAHs (6.3, 12.5, 25, 50, and 100 µg/mL) for 24 h. The results demonstrate that an increase in PAH concentration resulted in a decrease in cell viability in a dose-dependent manner (Figure 2). PAHs had no cytotoxic effects at concentrations lower than 12.5 µg/mL compared to control cells that received no treatment. However, cell viability began to decrease to below 80% when the PAH concentration increased to 25 µg/mL. Pan et al. demonstrated that PM, primarily composed of PAHs, exhibited higher cytotoxicity. Moreover, the barrier impairments from the PM primarily consisting of PAHS were significantly more severe than those primarily containing heavy metals [36]. Previous studies have also highlighted the enhanced toxic effects of PAH mixtures over individual PAHs, suggesting a potential synergistic interaction among different PAHs that exacerbates cellular damage and inflammatory responses [37,38].

To study the effects of CCS extract, CGA, CAF, and THP on PAH-induced cell toxicity, a dose-dependent toxicity test was conducted to select the optimal concentration of PAHs (12.5 µg/mL) and the tested compounds (62.5, 125, 250, 500, and 1000 µg/mL) for 24 h. The results showed that CCS extract (Figure 3a), CAF (Figure 3c), and THP (Figure 3d) exhibited no cytotoxicity in RAW 264.7 cells at concentrations below 125 µg/mL. In contrast, CGA (Figure 3b) exhibited cytotoxicity at concentrations above 62.5 µg/mL. However, a linear regression was not observed in the viability of CGA-treated cells. Therefore, all tested compounds were used in subsequent experiments at concentrations lower than 125 µg/mL. Interestingly, CGA demonstrated more cytotoxicity in RAW 264.7 cells compared to CAF and THP. This increased cytotoxicity may be due to its pro-oxidant activity at higher concentrations, leading to excessive ROS generation and oxidative stress [39,40].

#### 2.3.2. Effect of Inhibiting Intracellular Reactive Oxygen Species Production

Oxidative stress results from an imbalance between the generation of ROS and the protective action of antioxidants. When PM penetrates the skin, ROS production is a primary mechanism responsible for its negative impacts [41]. To evaluate the impact of PAH-induced oxidative stress, flow cytometry was used to measure intracellular ROS production in RAW 264.7 cells. ROS generation was detected using H_2_DCFDA, a redox-sensitive dye that fluoresces when oxidized by ROS. The ROS level in PAH-exposed cells was significantly elevated at a concentration of 12.5 μg/mL compared to untreated cells (Figure 4a), which corresponds to histograms of RAW 264.7 cell counts versus fluorescence intensity (Figure 4b).

However, treatment with CCS extract, CGA, CAF, and THP at 100 μg/mL inhibited the increase in ROS production induced by PAHs. PAHs increased the fluorescence intensity due to the oxidation of H2DCFDA, and this strong change was reduced by CCS extract, CGA, CAF, and THP compared to untreated cells. These results are consistent with the findings of Gualtieri et al., who reported a positive association between PAHs in PM and ROS production—the increased ROS levels induced by PAHs accelerated cell apoptosis and death [42]. Consequently, exposure to PM increases oxidative stress by promoting the generation of ROS, leading to inflammation through the release of pro-inflammatory cytokines such as TNF-α, IL-1β, iNOS, prostaglandin E2 (PGE2), and COX-2 [13]. The heightened production of these pro-inflammatory cytokines through the creation of ROS worsens the progression of inflammatory skin conditions, such as atopic dermatitis (A.D.), acne, and psoriasis. Additionally, skin aging is directly linked to the presence of ambient PM [43]. CAF effectively inhibits lipid peroxidation in rat liver microsomes, with the strongest effect against hydroxyl radicals, a moderate effect against singlet oxygen, and a lower effect against peroxyl radicals. Comparable to glutathione and superior to ascorbic acid, CAF’s ROS-quenching ability contributes to its anti-inflammatory effects by reducing oxidative stress [44]. The ability of CGA to reduce cell death and ROS production in keratinocytes induced by PM10 has also been reported [45]. These findings suggest that CCS extract, with its main compounds CGA, CAF, and THP, can reduce PAH-induced oxidative stress by lowering ROS production. This antioxidant effect might be responsible for mitigating the inflammatory response. These results indicate the potential of CCS extract and its components as effective agents for protecting against oxidative stress and inflammation induced by environmental pollutants.

#### 2.3.3. The Effects on Inhibiting Nitric Oxide Production

Nitric oxide (NO) is a highly reactive radical that acts as a pro-inflammatory mediator in various cell types, including macrophages [46]. The production of NO is a notable characteristic of RAW 264.7 cells when exposed to inflammatory stimuli. Therefore, we investigated the effects of PAHs on NO production in the cells and the inhibitory potential of CCS extract compared to CGA, CAF, and THP to reduce NO production. The results demonstrated a dose-dependent increase in NO production, reaching the highest level at 12.5 μg/mL of PAHs compared to untreated cells (Figure 5). Specifically, at a concentration of 12.5 μg/mL, PAHs induced a three-fold increase in NO levels compared to the control group, indicating significant stimulation of NO secretion by RAW264.7 cells. Consequently, this concentration resulted in the highest NO production without cytotoxic effects. It was selected for subsequent experiments to ensure that the observed increase in nitric oxide was due to inflammatory processes rather than cell death.

To explore the potential of CCS extract in inhibiting NO production in RAW 264.7 cells exposed to PAHs, its effects were compared to the standard compounds CGA, CAF, and THP. As shown in Figure 6, PAHs significantly upregulated NO production, which was dose-dependently downregulated by treatment with CCS extract, CGA, THP, and CAF. Notably, at a concentration of 100 μg/mL, CCS extract exhibited a comparable ability to inhibit NO production to that of CGA and THP. CGA is known for its anti-inflammatory actions via suppression of NO expression [47]. THP also exhibits anti-inflammatory effects by inhibiting the L-arginine-dependent production of NO in peripheral blood mononuclear cells [48]. The similar effectiveness of CCS extract compared to these established compounds suggests a possible synergistic effect from its various components. This synergistic interaction might account for the potent inhibitory effect of CCS extract on NO production observed in this study.

#### 2.3.4. Effect on Inhibiting Secretion and Gene Expression of Pro-Inflammatory Mediators

Inflammatory cytokines, such as IL-6, TNF-*α*, iNOS, and COX-2, are biomarkers involved in the development of inflammatory diseases [49]. Various cell types, including macrophages, fibroblasts, and keratinocyte cells, release these cytokines, contributing to the complex inflammatory response observed in skin conditions [50]. Macrophages can be induced to secrete NO and a series of cytokines, including TNF-*α*, IL-6, and COX-2, which express NF-*κ*B-dependent iNOS and COX-2. The overproduction of these cytokines is implicated in inflammatory skin diseases [51]. Several studies have suggested that TNF-*α* and IL-6 play crucial roles in initiating and maintaining the inflammatory response, leading to the pathogenesis of conditions such as contact dermatitis and psoriasis [52,53,54]. IL-6 further contributes to this process by promoting T-cell differentiation and acute-phase protein synthesis, which enhances the inflammatory environment [55]. Therefore, inhibiting the synthesis of these cytokines could be effective in preventing or suppressing inflammatory disorders. However, the effect of CCS extract on preventing inflammation caused by PAHs, a primary toxic component of PM, has not been explored. The activation of TNF-*α* increases the release of inflammatory cytokines such as IL-1*β*, IL-6, and iNOS, which are important indicators for determining inflammation severity [56,57]. To find out how the PAHs affected inflammation, the levels of TNF-*α* and IL-6 in RAW 264.7 cells were measured using ELISA. PAHs at a concentration of 12.5 µg/mL were used to stimulate the cells, and the results revealed that the levels of TNF-*α* and IL-6 significantly increased in RAW 264.7 cells, as shown in Figure 7a and Figure 7b, respectively.

However, treatment with CCS extract, CGA, CAF, and THP significantly reduced the production of TNF-*α* in a concentration-dependent manner compared to PAHs alone. At 100 μg/mL, the CCS extract decreased TNF-*α* levels approximately 20-fold compared to PAH-induced cells, demonstrating an inhibitory effect similar to that of THP at the same concentration (Figure 8a). CGA also effectively decreased TNF-*α* production, and notably, its ability to reduce TNF-α levels at 100 µg/mL was similar to that of the CCS extract at the same concentration. While CAF was able to lower TNF-*α* levels, it was less potent, with TNF-*α* concentrations remaining higher than those observed with CCS extract, CGA, and THP at equivalent concentrations. A study by Lemos et al. showed that CGA and CAF, which are found in large amounts in Robusta coffee beans (50 μg/mL), had a strong ability to inhibit IL-6 and TNF-*α* in macrophage cultures that had been stimulated with lipopolysaccharide (LPS) [58]. CGA also reduces the levels of TNF-*α*, IL-6, and IL-1*β* in rats with acetaminophen-induced liver damage [59]. Furthermore, rats administered CAF exhibited a reduction in IL-6, TNF-*α*, and cellular apoptosis compared to rats treated with a placebo, demonstrating the efficacy of CAF in improving biochemical indicators in models of liver injury [60]. Ezeamuzie et al. reported that THP (50 μM) significantly inhibited the release of TNF-*α* and IL-6 from human monocytes, comparable with dexamethasone (0.1 μM), indicating that THP has a strong anti-inflammatory effect similar to that of dexamethasone [61]. These results indicated that CCS extract and its effective compounds, CGA, CAF, and THP, exhibit significant anti-inflammatory activity by downregulating the pro-inflammatory cytokines TNF-*α* and IL-6. The presence of these substances in CCS extract is expected to result in synergistic effects, consequently improving the overall anti-inflammatory efficacy. The CCS extract is a powerful anti-inflammatory mediator since each phytochemical helps regulate important inflammatory mediators.

Inducible nitric oxide synthase (iNOS) and COX-2 are enzymes crucial to the inflammatory response. iNOS enhances NO production during inflammation, while COX-2 converts arachidonic acid to prostaglandins, key players in the inflammatory process [62,63]. Elevated expression of these genes is often observed in inflammatory skin diseases, making them significant targets for treatment [64]. Increased NO production, driven by iNOS and COX-2 activation, is implicated in the development of several inflammatory conditions. To assess the impact of CCS extract and its main constituents, CGA, CAF, and THP, on the expression of these inflammatory markers, RT-PCR analysis was conducted on RAW 264.7 cells exposed to PAHs. Gene expression levels were normalized to the control group, with baseline expression set to 1.

As shown in Figure 9a, PAH exposure significantly upregulated iNOS mRNA expression compared to untreated cells. However, treatment with CCS extract, CGA, CAF, and THP inhibited iNOS expression. Remarkably, CCS extract at 100 μg/mL significantly reduced NO production and iNOS expression, comparable to CGA, CAF, and THP. Similarly, PAHs significantly upregulated COX-2 mRNA expression (Figure 9b). Treatment with CCS extract, CGA, CAF, or THP can decrease COX-2 gene expression. Specifically, CCS extract at 100 μg/mL markedly inhibited COX-2 mRNA expression comparable to CGA, CAF, and THP. Previous studies by Cao et al. suggested that CGA strongly inhibits NF-*κ*B and MAPKs, which are crucial for the transcription of inflammatory genes, including COX-2 and iNOS [65]. THP’s inhibition of N.F.-κB leads to decreased production of various pro-inflammatory cytokines and mediators, highlighting its anti-inflammatory potential [66]. CAF significantly reduces LPS-induced inflammation in RAW 264.7 cells by decreasing NO production and downregulating pro-inflammatory genes such as iNOS, COX-2, IL-3, IL-6, and IL-12 [67]. Experiments using a zebrafish model further confirmed CAF’s anti-inflammatory effects, showing suppressed LPS-induced NO production [68].

The molecular inflammatory process also plays a significant role in aging and age-related diseases, where COX-derived reactive species and the transcriptional activity of IL-1*β*, IL-6, TNF-*α*, COX-2, and iNOS are increased [69,70]. Ontawong et al. reported that coffee pulp extract, containing CGA and CAF, suppresses inflammatory cytokines and mediators (TNF-*α*, IL-6, IL-1*β*, COX-2, iNOS, NO, and PGE_2_) in LPS-activated murine macrophage cells by inactivating the NF-*κ*B and MAPK signaling pathways, indicating CAF and CGA may contribute to the anti-inflammatory effects of the extract [71]. Furthermore, treatment with CCS extract contributes to marked decreases in the inflammation severity induced by TNF-*α*, likely via NF-*κ*B signaling, which acts downstream of TNF-*α* [72]. These data suggest that CCS extract may be used to fight PAH-induced skin inflammation by suppressing the expression of iNOS and COX-2. The anti-inflammatory properties of CCS extract are mainly probably attributed to its constituents, specifically CGA, CAF, and THP. These compounds have the potential to alleviate skin conditions such as psoriasis and aging.

### 2.4. Irritation Assessment of Coffee Cherry Pulp Extract Using Hen’s Egg Test on the Chorioallantoic Membrane (HET-CAM) Assay

The hen’s egg test–chorioallantoic membrane (HET-CAM) test is an acceptable substitute for the Draize rabbit eye test for evaluating the irritant properties of test chemicals [73]. In this study, the HET-CAM test was employed to examine the safety of CCS extract. As shown in Figure 10, the positive control (1% *w*/*v* SLS) caused immediate and severe indications of irritation on the CAM, including hemorrhage, coagulation, and vascular lysis. The irritation became even more acute after 60 min with an irritation score (I.S.) of 11.9 ± 0.6, indicating a strong irritating classification. Conversely, the negative control (0.9% *w*/*v* NaCl) did not cause any irritation (I.S. = 0). No signs of irritation were observed on the CAM during a 60 min application of CCS extract, indicating its safety for topical use. These results are consistent with other studies that have demonstrated the non-irritating nature of natural extracts containing CGA when tested using the HET-CAM method [74]. The findings support the safety of using CCS extracts for topical treatments, suggesting their potential as safe ingredients in cosmetic and therapeutic applications.

## 3. Materials and Methods

### 3.1. Materials

The standard reference material, PAH (CRM47930, QTM PAH-Mix, 2000 μg/mL), was purchased from Sigma-Aldrich (St. Louis, MO, USA). 6-hydroxy-2,5,7,8-tetramethyl chroman-2-carboxylic acid (Trolox), 2,2-azino-bis(3-ethylbenzthiazoline)-6-sulfonic acid (ABTS), 2,4,6-Tripyridyl-S-triazine (TPTZ), ferrous chloride (FeCl_2_), ferric chloride (FeCl_3_), ferrous sulfate (FeSO_4_), bovine serum albumin (BSA), soy lipoxygenase enzyme (652,799 units/mg), tris hydrochloride (Tris-HCL), CGA, CAF, THP, 3-(4,5-dimethylthiazolyl-2)-2,5-diphenyl tetrazolium bromide (MTT), and 2,7-dichlorodihydrofluorescein diacetate (H_2_DCFDA) were purchased from Sigma-Aldrich (St. Louis, MO, USA). Dulbecco’s Modified Eagle Medium (DMEM), fetal bovine serum (FBS), and penicillin-streptomycin were obtained from Gibco Life Technology (Thermo Fisher Scientific, Waltham, MA, USA). Ethanol and dimethyl sulfoxide (DMSO) were purchased from Labscan Asia Co., Ltd. (Bangkok, Thailand).

### 3.2. Preparation of Coffee Cherry Pulp Extract

Coffee cherry pulp (*Coffea arabica* L.) was collected from Doi Chang, Chiang Rai Province, in December. The coffee pulps were thoroughly cleansed, and coffee beans were removed from the pulp using a mechanical separator. The pulps were dried in a hot air oven at 50 ± 2 °C until pulverized into powder. The coffee cherry pulp extract was produced using the Soxhlet extraction method, as detailed in our previous study [14]. Briefly, the extraction was carried out using 95% ethanol with a solid-to-liquid ratio of 1:5 g/mL. To obtain coffee cherry pulp (CCS) extract, an extraction process using the Soxhlet apparatus was carried out for 60 min. The crude extract was stored in an airtight container and kept at a temperature of 4–8 °C before use.

### 3.3. Determination of Antioxidant Activities

#### 3.3.1. ABTS Free Radical Scavenging Assay

CCS extract, CGA, CAF, THP, and Trolox (positive control) were evaluated for free radical scavenging activity using an ABTS scavenging assay [75]. Briefly, ABTS radical cation (ABTS·+) was generated by mixing 7 mM of ABTS in ethanol with 2.45 mM potassium persulfate, then stored in the dark at room temperature for 24 h before use. Then, 180 µL of 1000 µg/mL of each tested compound was mixed with 20 µL of ABTS solution diluted in absolute ethanol. The mixture was incubated at room temperature in the dark for 15 min. The absorbance was determined with a UV-Vis spectrophotometer microplate reader (SpectraMax M3, San Jose, CA, USA) at a wavelength of 734 nm. A standard curve was plotted using the percentage inhibition of absorbance against Trolox concentration (0–100 µM). The percentage inhibition of each tested compounds, calculated using Equation (1), was compared to the Trolox standard curve to determine the equivalent Trolox concentration (µM). The antioxidant capacity was expressed as Trolox equivalent antioxidant capacity (TEAC) in µM Trolox equivalents.
% Inhibition = [(A − B)/A] × 100%(1)
where A is the absorbance of the ABTS·+ solution without the tested compounds. B is the absorbance with the tested compounds.

#### 3.3.2. Ferric Reducing Antioxidant Power (FRAP) Assay

The FRAP assay was used to assess the reducing capacity of CCS extract compared to CGA, CAF, and THP, following the method of Chiangnoon et al. [75]. Briefly, the FRAP reagent was prepared from a 10:1:1 mixture of 0.3 M acetate buffer (pH 3.6), 10 Mm of TPTZ dissolved in 40 mM of 37% HCl, and 20 mM ferric chloride solution. All tested compounds were diluted in absolute ethanol to a concentration of 1 mg/mL. In total, 20 µL of the tested compounds was added to a 96-well plate with 180 µL of FRAP reagent and incubated at 37 °C for 30 min. A microplate reader (SpectraMax M3, San Jose, CA, USA) was used to measure at 595 nm. As a standard, ferrous sulfate (100–1000 µM) was employed to calculate the FRAP value, which was expressed as mM FeSO_4_ equivalent.

### 3.4. Determination of Anti-Inflammatory Activities

#### 3.4.1. Protease Inhibition Assay

The ability of CCS extract, CGA, CAF, and THP to inhibit protease enzyme was evaluated following some modifications from Assiry et al. [76]. Diclofenac sodium was used as a positive control. The reaction mixture consisted of 250 μL of trypsin, 1 mL of 25 mM Tris-HCl buffer (pH 7.4) and 1 mL of tested compounds dissolved in DI water. After incubating at 37 °C for 5 min, 1.0 mL of 0.8% (*w*/*v*) BSA was added. The mixture was incubated for a further 20 min. To terminate the reaction, 2.0 mL of 70% (*v*/*v*) perchloric acid was added. The turbid suspension was centrifuged, and the absorbance of the supernatant was measured at 280 nm using a microplate reader (SpectraMax M3, San Jose, CA, USA). The absorbance was calculated to express the percentage inhibition of protease activity as follows:% Inhibition of Protease activity = [(A − B)/A] × 100%(2)
where A is the absorbance of the reaction with buffer, trypsin, BSA and perchloric acid. B is the absorbance of the reaction with extract, trypsin, BSA and perchloric acid.

#### 3.4.2. Lipoxygenase (LOX) Inhibition Assay

The LOX inhibitory action was assessed using the ferric oxidation of xylenol orange (FOX assay) according to some modified method of Chung et al. [77]. Briefly, CCS extract, CGA, CAF, THP and diclofenac sodium (positive control) were dissolved in DI water to a concentration of 1 mg/mL. In total, 100 ng/mL of lipoxygenase enzyme was dissolved in 50 mM Tris-HCL buffer (pH 7.4). Then, 20 µL of the tested compounds was added to 50 µL of enzyme solution, and the mixture was incubated at room temperature in the dark for 5 min. The reaction was initiated by adding 50 µL of 616 µM linoleic acid into the mixture. After 20 min in the dark, the reaction was terminated by adding 100 µL of freshly prepared FOX reagent (15 µM xylenol orange and 15 µM FeSO_4_ dissolved in a mixture of 15 mL of 300 mM H_2_SO_4_ and 135 mL of methanol). After 30 min of dark incubation, the absorbance of the mixture was measured at 560 nm using a microplate reader (SpectraMax M3, San Jose, CA, USA). The absorbance was calculated to express the percentage inhibition of lipoxygenase activity as follows:% Inhibition of LOX activity = [(A − B)/A] × 100%(3)
where A is the absorbance of the reaction with DI water, enzyme, linoleic acid, FOX reagent. B is the absorbance of the reaction with extract, enzyme, linoleic acid, FOX reagent.

### 3.5. Cell Culture

The macrophage cell line (RAW 264.7 cells, TIB-71; ATCC, Manassas, VA, USA) was cultured in Dulbecco’s Modified Eagle Medium (DMEM) with 10% fetal bovine serum (FBS) and 1% penicillin-streptomycin. The cells were maintained at 37 °C in a humidified incubator with 5% CO_2_ (Eppendorf, CellXpert C170, Hauppauge, NY, USA). In this study, a standard reference material of mixed PAHs (CRM 47930) was utilized to induce toxicity and promote ROS formation in the macrophage cells.

### 3.6. Cell Viability Assay

The MTT assay was employed to evaluate the cell viability of tested compounds exposed to PAHs by measuring the metabolic activity that converts MTT into formazan crystals [78]. RAW 264.7 cells were seeded at a density of 1× 10^5^ cells/well in 96-well plates. After a 24 h incubation period, the cells were pre-treated with varying concentrations of CCS extract, CGA, CAF, and THP (ranging from 0.002 to 1 mg/mL) for 2 h. This was followed by exposure to a PAH solution dissolved in serum-free DMEM (12.5 µg/mL) for 24 h. Subsequently, 0.5 mg/mL of MTT was added to the culture medium and incubated for 2 h. The medium was then removed, and formazan crystals were dissolved by adding 100 µL of DMSO. Cell viability was assessed by measuring the absorbance at 550 nm using a microplate reader (SpectraMax M3, San Jose, CA, USA). The optimal concentration that resulted in more than 80% cell survival was selected for further studies. The percentage of RAW 264.7 cell viability was calculated as follows:Cell viability (%) = [A/B] × 100%(4)
where A is the absorbance of the cells treated with extract. B is the absorbance of the cells treated with an incomplete medium.

### 3.7. Reactive Oxygen Species Production Inhibition (H_2_DCFDA Assay)

To evaluate the intracellular ROS scavenging ability of CCS extract and its main compounds, the ROS level was quantified using probe H_2_DCFDA [14]. RAW 264.7 cells were seeded at a density of 5 × 10^5^ cells/mL in 96-well plates and cultured for 24 h. The cells were pretreated with the tested compounds for 2 h prior to exposure to PAH solution (12.5 µg/mL) for 2 h. Following washing with PBS, a 2 µM H_2_DCFDA solution was added for further incubation of 15 min. The extract solution was centrifuged, and the obtained supernatant was analyzed for fluorescence intensity using a flow cytometer (BD Accuri™ C6 Plus, Franklin Lake, NJ, USA). The relative fluorescence intensity was calculated in comparison to the untreated control.

### 3.8. Intracellular Nitric Oxide Production

The anti-inflammatory effect of CCS extract and its main compounds was investigated via nitric oxide (NO) secretion from macrophage cells [79]. Briefly, RAW 264.7 cells were seeded at a concentration of 1 × 10^5^ cells/mL in a 96-well plate and incubated for 24 h. The cells were pretreated with the tested compounds for 2 h before exposure to PAH solution (12.5 µg/mL) for 24 h. After incubation, 100 µL of each supernatant solution was added to a 96-well plate, followed by a reaction with 100 µL of Griess reagent. The absorbance was then measured at 540 nm using a microplate reader (SpectraMax M3, San Jose, CA, USA), and the nitrite content was calculated from a standard curve of sodium nitrite.

### 3.9. Enzyme-Linked Immunoassay (ELISA) Analysis

The anti-inflammatory effects of the tested compounds were evaluated by measuring the production and release of the pro-inflammatory cytokines IL-6 and TNF-α. Briefly, RAW 264.7 cells were seeded at a concentration of 1 × 10^5^ cells/mL in a 96-well plate and incubated for 24 h. The cells were pretreated with the tested compounds for 2 h before exposure to PAH solution (12.5 µg/mL) for 24 h. After incubation, concentrations of IL-6 and TNF-α in the cell culture supernatants were quantified using ELISA MAX™ Deluxe kits for human IL-6 and TNF-α (BioLegend) according to the manufacturer’s instructions.

### 3.10. Semi-Quantitative Reverse Transcription and Polymerase Chain Reaction Analysis (RT-PCR)

The expression levels of inflammatory cytokines, including IL-6 and TNF-α, were evaluated using semi-quantitative RT-PCR. Briefly, RAW 264.7 cells were seeded at a concentration of 1 × 10^6^ cells/mL and cultured for 24 h. The cells were then pre-treated with different concentrations of the tested compounds for 20 h, followed by stimulation with a PAH solution (12.5 µg/mL) for 4 h. The cells were subsequently harvested for mRNA expression analysis. Total RNA was isolated using an RNA extraction kit (G.E. Healthcare, Chalfont St. Giles, UK), and the concentration was measured with a NanoDrop spectrophotometer (Thermo Fisher Scientific, Waltham, MA, USA). For semi-quantitative RT-PCR, RNA was reverse transcribed to cDNA using the Omniscript RT Kit (QIAGEN, Hilden, Germany). Each reaction consisted of 3 μL of cDNA and the specific primers listed in Table 2. The cDNA was then amplified using β-actin as the internal reference. Amplification was performed in a thermal cycler (A.B. Applied Biosystems GeneAmp PCR System 2400) under the following conditions: an initial denaturation step at 95 °C for 2 min, followed by 27 cycles of denaturation at 95 °C for 15 s, annealing at 60 °C for 60 s, and extension at 72 °C for 60 s.

### 3.11. Irritation Property by Hen’s Egg Test on the Chorioallantoic Membrane (HET-CAM) Assay

The irritation potential of CCS extract was assessed using the hen’s egg test–chorioallantoic membrane (HET-CAM) assay [80]. Sample solutions were added to the chorioallantoic membrane (CAM) and observed for 5 min. The times at which signs of irritation, such as vascular bleeding, lysis, and coagulation, first appeared were precisely recorded. The severity of irritation was graded based on both the rapidity and intensity of these signs. Normal saline solution was used as the negative control, while 1% *w*/*v* sodium lauryl sulfate served as the positive control. The irritation score (I.S.) was calculated as follows:I.S. = [(301 − t(h))/300 × 5] + [(301 − t(l))/300 × 7] + [(301 − t(c))/300 × 9](5)
where t(h) is the time (s) that the first vascular hemorrhages occurred. t(l) is the time (s) that the first vascular lysis occurred, and t(c) is the time (s) that the first vascular coagulation occurred. The results were classified as no irritation (I.S. = 0.0–0.9), slight irritation (I.S. = 1.0–4.9), moderate irritation (I.S. = 5.0–8.9), and severe irritation (I.S. = 9.0–21.0). Photographs of the blood vessels on CAM were taken under a stereo microscope.

### 3.12. Statistical Analysis

Data were presented as mean ± S.D. Each experiment was conducted in triplicate per sample. Multiple comparisons among the various treatment groups were achieved using one-way analysis of variance (ANOVA), followed by Tukey’s test for post hoc analysis. A significance level of *p* < 0.05 was considered.

## 4. Conclusions

The anti-inflammatory effect of CCS extract against atmospheric PAH exposure was demonstrated for the first time in this study. CCS extract, along with its key constituents, CGA, CAF, and THP, significantly reduced the production of pro-inflammatory cytokines and mediators such as TNF-α, IL-6, iNOS, and COX-2 in RAW 264.7 cells. These indicate its potential to mitigate the inflammatory responses induced by PAHs. Furthermore, CCS extract effectively reduced ROS production and NO levels and inhibited inflammatory enzymes, contributing to its overall anti-inflammatory and antioxidant properties. The HET-CAM test confirmed the safety of CCS extract for topical application, showing no signs of irritation. These results suggest that CCS extract could be a valuable natural ingredient for developing anti-inflammatory cosmeceutical products aimed at protecting against pollution. Future studies should focus on in vivo assessments and the exploration of its effects on human skin to further substantiate its efficacy and safety. In conclusion, the comprehensive anti-inflammatory properties of CCS extract, supported by the synergistic effects of its bioactive compounds, position it as a promising candidate for mitigating air pollution-induced skin inflammation and associated skin disorders.

## Figures and Tables

**Figure 1 ijms-25-09416-f001:**
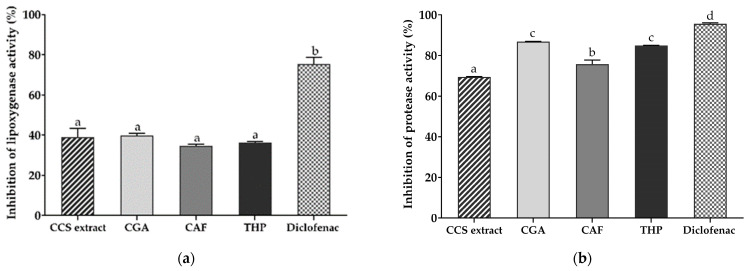
The ability of coffee cherry pulp (CCS) extract to inhibit (**a**) lipoxygenases (LOX) and (**b**) protease enzymes compared to a positive control (diclofenac sodium) and standard compounds chlorogenic acid (CGA), caffeine (CAF), and theophylline (THP), all tested at the same concentration of 1000 µg/mL. Data are presented as the mean ± S.D. (*n* = 3). Tukey’s HSD test was conducted to compare all group means. Groups sharing the same letters (a–d) are not significantly different, with a significance level of *p* < 0.05.

**Figure 2 ijms-25-09416-f002:**
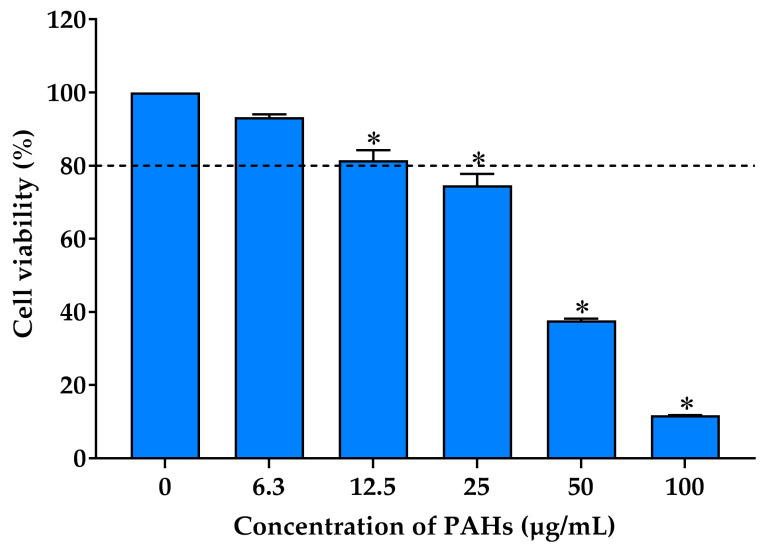
Effects of PAHs on the viability of RAW 264.7 cells using the MTT assay; the bars represent the percent cell viability of the cells treated with different concentrations of the PAHs (6.3–100 µg/mL) for 24 h. Data are expressed as mean ± S.D. (*n* = 3). An asterisk (*) indicates a significant difference compared to the untreated cells (0 µg/mL) at *p* < 0.05.

**Figure 3 ijms-25-09416-f003:**
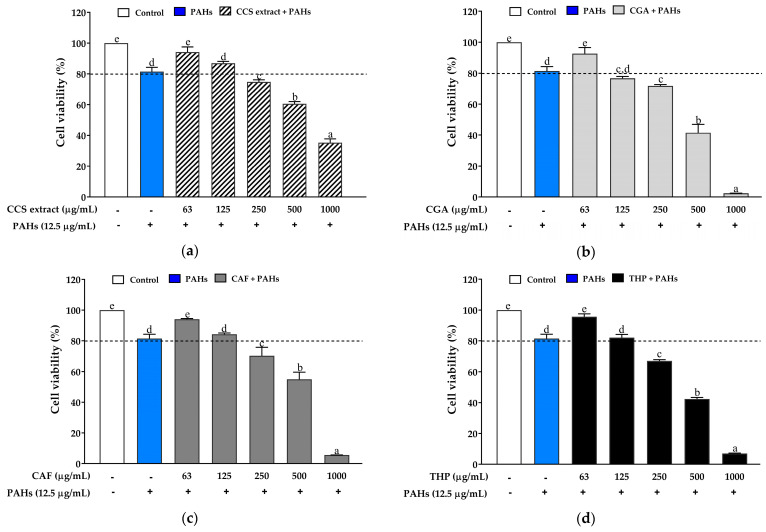
Effects of (**a**) coffee cherry pulp (CCS) extract, (**b**) chlorogenic acid (CGA), (**c**) caffeine (CAF), and (**d**) theophylline (THP) on the viability of RAW 264.7 cells using the MTT assay; the bars represent the percent cell viability of the cells treated with tested compounds in different concentrations (62.5–1000 µg/mL) then exposed to 12.5 µg/mL of PAHs for 24 h. Data are presented as the mean ± S.D. (*n* = 3). Tukey’s HSD test was conducted to compare all group means. Groups sharing the same letters (a–e) are not significantly different, with a significance level of *p* < 0.05.

**Figure 4 ijms-25-09416-f004:**
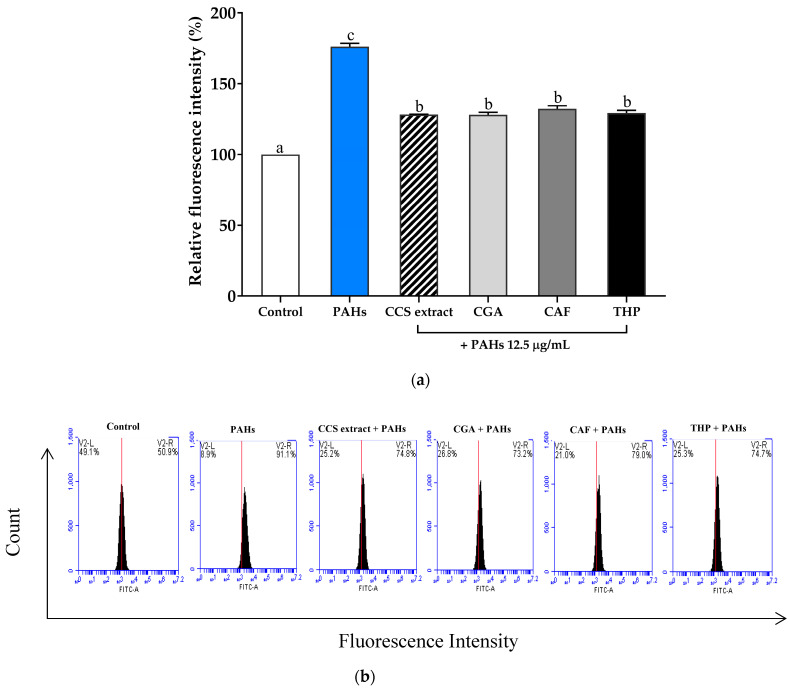
(**a**) Effect of coffee cherry pulp (CCS) extract, chlorogenic acid (CGA), caffeine (CAF), and theophylline (THP) at the same concentration of 100 μg/mL on ROS production in RAW 264.7 cells exposed to PAHs. The cells were pre-treated with tested compounds and PAHs (12.5 μg/mL) for 2 h to determine ROS production. Data are presented as the mean ± S.D. (*n* = 3). Tukey’s HSD test was conducted to compare all group means. Groups sharing the same letters (a–c) are not significantly different, with a significance level of *p* < 0.05. (**b**) Histograms of the RAW 264.7 cell counts versus fluorescence intensity are shown with a mark to define fluorescing cells of the tested compounds (100 μg/mL) alone or in combination with PAHs.

**Figure 5 ijms-25-09416-f005:**
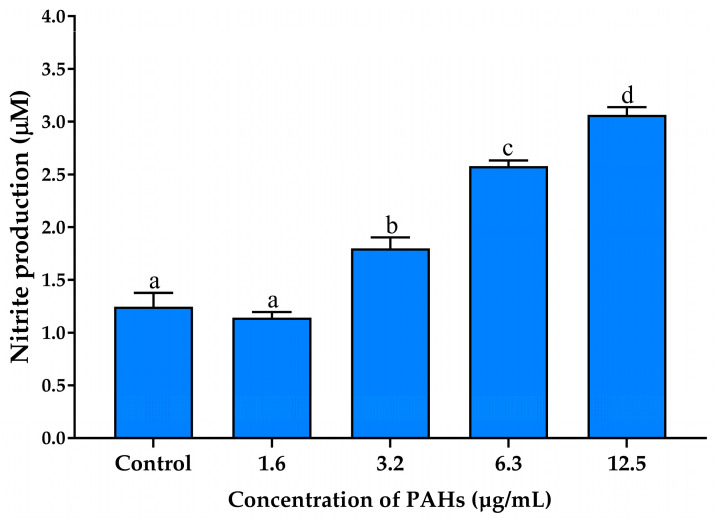
Effects of PAHs on NO production in RAW 264.7 cells; the bars represent the nitric oxide production of the cells treated with different concentrations of the PAHs (1.6–12.5 µg/mL) for 24 h. Data are presented as the mean ± S.D. (*n* = 3). Tukey’s HSD test was conducted to compare all group means. Groups sharing the same letters (a–d) are not significantly different, with a significance level of *p* < 0.05.

**Figure 6 ijms-25-09416-f006:**
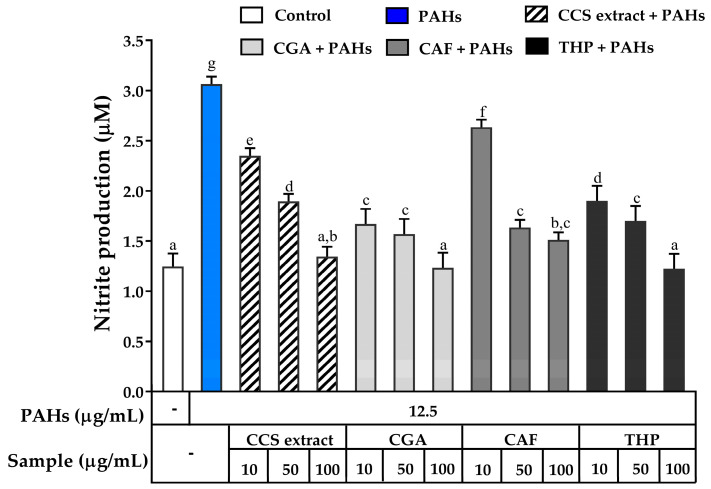
Inhibitory effects of coffee cherry pulp (CCS) extract, chlorogenic acid (CGA), caffeine (CAF), and theophylline (THP) on PAH-induced production of nitric oxide. RAW 264.7 cells were pre-treated with various concentrations (10, 50, and 100 μg/mL) of the tested compounds for 2 h and then incubated with PAHs (12.5 μg/mL) for 22 h. Data are presented as the mean ± S.D. (*n* = 3). Tukey’s HSD test was conducted to compare all group means. Groups sharing the same letters (a–g) are not significantly different, with a significance level of *p* < 0.05.

**Figure 7 ijms-25-09416-f007:**
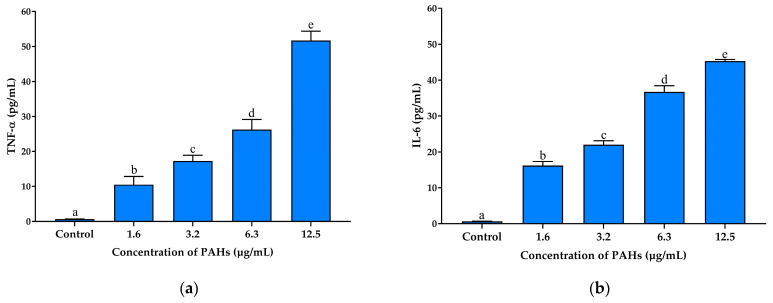
Effects of PAHs on TNF-*α* and IL-6 secretion in RAW 264.7 cells; the bars represent (**a**) the TNF-*α* and (**b**) IL-6 production of the cells treated with different concentrations of the PAHs (1.6–12.5 µg/mL) for 24 h. Data are presented as the mean ± S.D. (*n* = 3). Tukey’s HSD test was conducted to compare all group means. Groups sharing the same letters (a–e) are not significantly different, with a significance level of *p* < 0.05.

**Figure 8 ijms-25-09416-f008:**
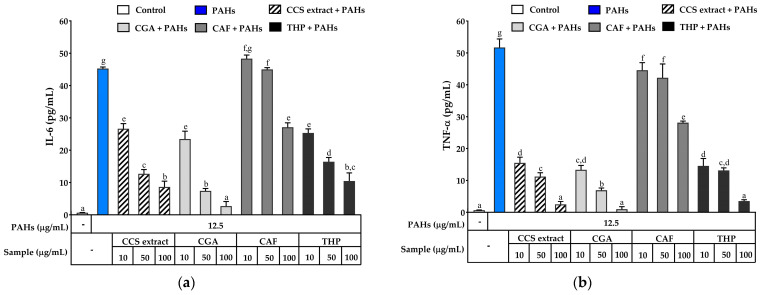
Inhibitory effects of coffee cherry pulp (CCS) extract, chlorogenic acid (CGA), caffeine (CAF), and theophylline (THP) on PAH-induced production of (**a**) TNF-*α* and (**b**) IL-6 in RAW 264.7 cells at 24 h. The cells were pre-treated with various concentrations (10, 50 and 100 μg/mL) of the tested compounds for 2 h and then incubated with PAHs (12.5 μg/mL) for 22 h. Data are presented as the mean ± S.D. (*n* = 3). Tukey’s HSD test was conducted to compare all group means. Groups sharing the same letters (a–g) are not significantly different, with a significance level of *p* < 0.05.

**Figure 9 ijms-25-09416-f009:**
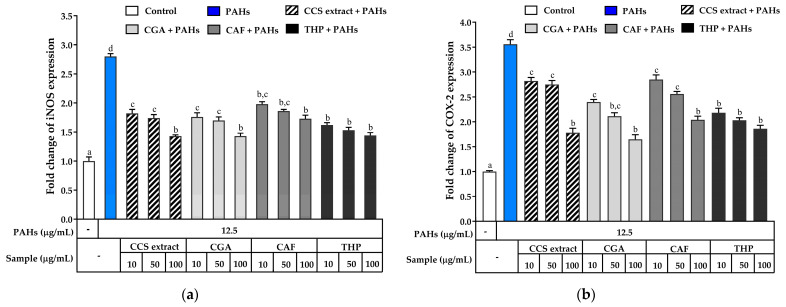
Inhibitory effects of coffee cherry pulp (CCS) extract, chlorogenic acid (CGA), caffeine (CAF), and theophylline (THP) on PAH-induced mRNA expression of (**a**) COX-2 and (**b**) iNOS. RAW 264.7 cells were pre-treated with various concentrations (10, 50, and 100 μg/mL) of the tested compounds for 2 h and then incubated with PAHs (12.5 μg/mL) for 22 h. Data are presented as the mean ± S.D. (*n* = 3). Tukey’s HSD test was conducted to compare all group means. Groups sharing the same letters (a–d) are not significantly different, with a significance level of *p* < 0.05.

**Figure 10 ijms-25-09416-f010:**
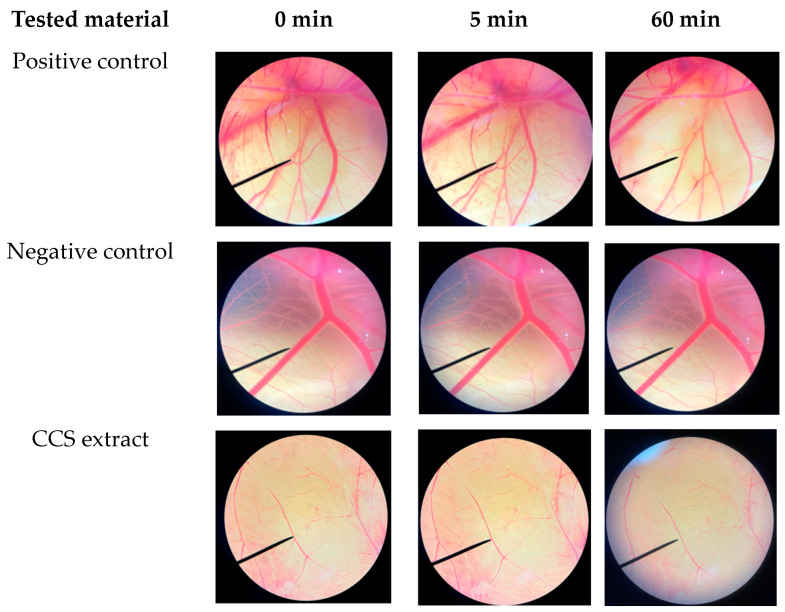
CAM was exposed to a positive control (1% *w*/*v* SLS), a negative control (0.9% *w*/*v* NaCl), and coffee cherry pulp (CCS) extract (10 mg/mL) for 5 and 60 min, respectively.

**Table 1 ijms-25-09416-t001:** Antioxidant activities (ABTS and FRAP assays) of the coffee cherry pulp extract compared to chlorogenic acid, caffeine, and theophylline tested at the same concentration of 1000 µg/mL.

Tested Compound	Antioxidant Activity	
	ABTS TEAC (µM Trolox/g)	FRAP (mM FeSO_4_)
CCS extract	193.6 ± 3.7 ^a^	6.52 ± 0.7 ^a^
CGA	310.3 ± 2.5 ^d^	18.71 ± 1.3 ^d^
CAF	213.8 ± 1.0 ^c^	10.05 ± 0.9 ^c^
THP	204.6 ± 1.4 ^b^	8.30 ± 1.0 ^b^

CCS extract = Coffee cherry pulp extract; CGA = Chlorogenic acid; CAF = Caffeine; THP = Theophylline. The values are expressed as the mean ± standard deviation (S.D.) for three replicates (*n* = 3). Within the same column, different letters indicate statistically significant differences between tested compounds, with a significance level of *p* < 0.05, determined using Tukey’s Honest Significant Difference (HSD) test following a one-way ANOVA.

**Table 2 ijms-25-09416-t002:** Sequences of gene-specific primers used for semi-quantitative RT-PCR.

Gene	Primer	Sequence: (5′-3′)
β–actin	Forward	TCATGAAGTGTGACGTTGACATCCGT
	Reverse	CCTAGAAGCATTTGCGGTGCACGATG
IL-6	Forward	CATCCAGTTGCCTTCTTGGGA
	Reverse	GCATTGGAAATTGGGGTAGGAAG
TNF-α	Forward	ATGAGCACAGAAAGCATGATC
	Reverse	TACAGGCTTGTCACTCGAATT

## Data Availability

Data are contained within this article.

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
