# Peer review of "Potential of Coffee Cherry Pulp Extract against Polycyclic Aromatic Hydrocarbons in Air Pollution Induced Inflammation and Oxidative Stress for Topical Applications"

_ijms, 2024, doi:10.3390/ijms25179416_

Round 1

Reviewer 1 Report

Comments and Suggestions for Authors

The manuscript deals with the potential use of coffee cherry pulp (CCS), a by-product of coffee bean harvesting, that is considered waste. The three main compounds are chlorogenic acid, caffeine and theophylline known for their antioxidant, anti-inflammatory, anti-ageing properties. Therefore, CCS extract has significant potential for various health benefits.

The authors first report the in vitro antioxidant potential by the ABTS and DPPH methods of the CCS extract and the pure components CGA, CAF, and THP and then, they evaluated: the lipoxygenase and protease inhibitory effects, anti-inflammatory effect against polycyclic aromatic hydrocarbons (PAHs).  Suppression of other inflammatory factors such as NO, TNF-α, IL-6, iNOS and COX-2 was evaluated. The HET-CAM test was used to examine the safety of the CCS extract.

The manuscript is well-structured, and the arguments appear well supported by experiments and references. The novelty and originality of the manuscript are good and there are interesting data to contribute to the anti-inflammatory properties of CCS extract. The results are well discussed.  However, there are aspects that need to be clarified in the text.

- Line 187 shows the optimal concentrations of the samples used (63, 125…. ug/ml). Check Fig 3 (a,b,c,d) which shows the concentration of 62.5 ug/ml and Legend of Figure 3 (63 ug/ml)

Comments on the Quality of English Language

Minor editing of English language required

Author Response

Dear Reviewer of International Journal of Molecular Sciences,

Title: Potential of Coffee Cherry Pulp Extract Against Polycyclic Aromatic Hydrocarbons in Air Pollution Induced Inflammation and Oxidative Stress for Topical Applications (Manuscript ID: ijms-3137443)

We greatly appreciate the valuable comments and suggestions provided by all reviewers. We have carefully read and responded each comment, responding to them point by point. The specific alterations in the manuscript in response to the reviewer comments are shown in yellow highlight for comments of reviewer 1.

We hope that all of the changes have adequately addressed the reviewers’ concerns, so with these improvements, we sincerely hope our work will be accepted for publication in the International Journal of Molecular Sciences.

Sincerely yours,

Kanokwan Kiattisin

Department of Pharmaceutical Sciences

Faculty of Pharmacy, Chiang Mai University, Chiang Mai 50200 Thailand

E-mail: kanokwan.k@cmu.ac.th

Reviewer 2 Report

Comments and Suggestions for Authors

The manuscript entitled " Potential of Coffee Cherry Pulp Extract Against Polycyclic Aromatic

Hydrocarbons in Air Pollution Induced Inflammation and Oxidative Stress for Topical Applications” by Weeraya Preedalikit, Chuda Chittasupho, Pimporn Leelapornpisid, Natthachai Duangnin and Kanokwan Kiattisin, evaluates the antioxidant and anti-inflammatory activities of Coffee Cherry extract (CCS) on oxidative stress and inflammation induced by atmospheric polycyclic aromatic hydrocarbons in macrophage cells highlighting its potential use in the cosmeceutical field.

The manuscript is clearly produced in most of its sections and opens up the possibility that in the future CCS, a by-product of coffee bean harvesting, will not be considered a waste but recycled for the development of new health products.

However, I believe that the following minor revisions need to be made for publication.

Minor revisions

1)        Review in Line 24 the sentence “The CCS extract was assessed for its impact on the expression of nitric oxide, TNF-α, IL-6, iNOS, and COX-2” It is more correct to write …. the production of nitric oxide and expression of TNF-α, IL-6, iNOS, and COX-2 because nitric oxide is not a protein that is expressed 

2)        Check all acronyms included in the paper and to express them in the full form when first written in the manuscript (e.g. CAF and THP in line 64 and not in Line 101)

3)        In the table and in the figures please indicate more clearly the quantity of extract used and the composition in CGA, CAF e THF. Is it the same in every experiment?

4)        In Lines 172-174 please write a clearer sentence. What consists mainly of heavy metals?

5)        In Figure 3 please make sure that it is all (3a, 3b, 3c and 3d) on the same page

6)        Throughout the manuscript controls and improves paragraph separation (e.g. enter a space between Line 126  and Line 126, between  Line 281 and Line 282, etc.)

7)        Throughout the manuscript review the insertion of the term “samples” not used correctly (e.g. in Line 187 improve the sentence,  in Line 237 etc.)

8)        The paragraph 2.3.4.” Effect on Inhibiting Secretion and Gene Expression of Pro-Inflammatory Cytokines” please rewrite it better. Here the term “cytokines” is sometimes used improperly by also incorporating COX2 and iNOS. Please describe the results of Figure 8 more clearly (e.g. in Line 313 CCS does not have a similar effect to CAF and in Lines 316-318)

Kind Regards

Author Response

Dear Reviewer of International Journal of Molecular Sciences,

Title: Potential of Coffee Cherry Pulp Extract Against Polycyclic Aromatic Hydrocarbons in Air Pollution Induced Inflammation and Oxidative Stress for Topical Applications (Manuscript ID: ijms-3137443)

We greatly appreciate the valuable comments and suggestions provided by all reviewers. We have carefully read and responded each comment, responding to them point by point. The specific alterations in the manuscript in response to the reviewer comments are shown in green highlight for comments of reviewer 2. 

We hope that all of the changes have adequately addressed the reviewers’ concerns, so with these improvements, we sincerely hope our work will be accepted for publication in the International Journal of Molecular Sciences.

Sincerely yours,

Kanokwan Kiattisin

Department of Pharmaceutical Sciences

Faculty of Pharmacy, Chiang Mai University, Chiang Mai 50200 Thailand

E-mail: kanokwan.k@cmu.ac.th
